# The Development and Proof of Principle Test of TRIAGE: A Practical Question Set to Identify and Discuss Medication-Related Problems in Community Pharmacy

**DOI:** 10.3390/pharmacy8040178

**Published:** 2020-09-27

**Authors:** Marcia Vervloet, Hanneke E. Zwikker, Annemiek J. Linn, Ellen S. Koster, Suzan G. H. Gipmans, Maaike C. W. van Aarle, Liset van Dijk

**Affiliations:** 1Nivel, Netherlands Institute for Health Services Research, 3500 BN Utrecht, The Netherlands; l.vandijk@nivel.nl; 2Dutch Institute for Rational Use of Medicines, 3502 GB Utrecht, The Netherlands; h.zwikker@ivm.nl; 3Amsterdam School of Communication Research (ASCoR), University of Amsterdam, 1001 NG Amsterdam, The Netherlands; a.j.linn@uva.nl; 4Utrecht Pharmacy Practice Network for Education and Research (UPPER), Utrecht University, 3584 CG Utrecht, The Netherlands; e.koster@uu.nl; 5BENU Netherlands B.V., 3604 BB Maarssen, The Netherlands; s.gipmans@knmp.nl (S.G.H.G.); mvaarle@brocacef.nl (M.C.W.v.A.); 6Department of PharmacoTherapy, -Epidemiology & -Economics (PTEE), Groningen Research Institute of Pharmacy, Faculty of Mathematics and Natural Sciences, University of Groningen, 9700 AD Groningen, The Netherlands

**Keywords:** clinical pharmacy, community pharmacy, pharmaceutical care, patient-centered communication, patient-provider communication, medication experiences, medication-related problems, medication adherence, cardiovascular medication, practical question set

## Abstract

The pharmacy counter is a good place to identify and discuss medication-related problems. However, there is a lack of practical communication tools to support pharmacy technicians (PTs) in initiating a conversation with patients. This study aimed to develop and test a practical set of questions for PTs, called TRIAGE, to identify problems during encounters. TRIAGE was developed based on insight from the literature, focus groups with PTs and pharmacists, and input from patients and experts. In 10 community pharmacies, 17 PTs used TRIAGE during encounters with patients who collected their cardiovascular medication. For each encounter, PTs registered the identified problems and suggested solutions. A total of 105 TRIAGE conversations were held, 66 for first refill and 39 for follow-up refill prescriptions. In 15 (23%) first refill prescription encounters, a problem was identified. These problems concerned forgetting to take the medication, a complex medication regime or (fear of) side effects. In three (8%) follow-up refill prescription encounters, a problem was identified. Most of the problems were solved on the spot. Pharmacy technicians indicated that they identified medication-related problems with TRIAGE that otherwise would be left unnoticed. They appreciated TRIAGE as a useful instrument for starting the conversation with patients about medication use.

## 1. Introduction

Many patients experience medication-related problems varying from practical problems, e.g., problems with opening the package, a complex daily regimen, and forgetting the intake, to perceptual problems, e.g., fear of side effects, or other concerns about the medication, low necessity beliefs [1,2,3,4]. As a result, existing health problems may worsen or new health problems may arise [5,6].

The pharmacy is a good place to identify, discuss and possibly solve or prevent medication-related problems at an early stage, as patients visit their pharmacy regularly to collect their medication. In the Netherlands, a community pharmacy team comprises of pharmacy technicians (PTs) and pharmacists who supervise the PTs. Pharmacy technicians complete three years of vocational education after their secondary education and are the staff members who interact most with patients at the counter. In the last decade, the role of the pharmacy team has changed and shifted from preparing and handing out the medication to providing pharmaceutical patient care. Counseling patients about their medication, detecting and resolving potential medication-related problems, and supporting patients in their adherence to medication are core elements of pharmaceutical care aimed at improving medication use and health outcomes [7,8]. Moreover, the physician often has limited time to counsel patients on prescribed medication, providing opportunities for the pharmacy team [9]. However, previous research has shown that communication during pharmacy encounters can be improved [10,11,12,13,14]. Patients’ experiences, perceptions and preferences are discussed seldom at the pharmacy counter [10,11,15], while such discussions are needed to gain insights into problems that patients may encounter [16,17].

It is therefore important to support pharmacy staff in their communication with patients. Guidelines have been developed for this purpose, for example, the American Society of Health-System Pharmacists (ASHP) Guidelines on Pharmacist-Conducted Patient Education and Counseling [18] and the Pharmacy Guide on Counselling Patients on Medicines from the Royal Pharmaceutical Society [19]. In the Netherlands, the Royal Dutch Pharmacy Association (KNMP) has recently developed the guideline “Pharmaceutical Consultation” [20]. It is based on the Calgary–Cambridge model, which provides a structure in the conversation and helps build a trusting relationship with the patient [21,22]. According to this model, a pharmacy encounter should include four consecutive steps: (1) initiating the conversation; (2) gathering information; (3) explanation, advice and decision; and (4) closing the conversation. While most guidelines provide general information on the consultation process, they do not provide practical communication tools to optimally perform these consultations.

Several communication tools have already been developed. The most widely used tool appears to be the ‘Three Prime Questions’, developed by the US Indian Health Service [23]. While these questions are useful for verification of patients’ knowledge about the medication, they are less sensitive to identifying possible barriers that patients experience in their medication use. Another example of enhancing communication at the pharmacy counter was tested in the US Pennsylvania Project, which used the “Adherence Estimator™” screening instrument to identify problems with medication adherence [24]. However, this instrument and other similar instruments specifically aim to detect adherence problems and provide less room for asking patients about their medication experiences. This is important as previous research has shown that patients hardly initiate a conversation about medication-related problems at the pharmacy counter [10,11]. To facilitate an open conversation in which patients feel safe and encouraged to share their experiences with their medication, it is crucial that PTs ask the right questions. Moreover, when a medication-related problem is identified, the PT must go through a triage process: determining which follow-up actions should be initiated and by whom, e.g., the pharmacist, physician, practice nurse or another healthcare professional, if the identified problem is too complex for the PT to provide a solution.

Therefore, the aims of this study were: (1) to develop a practical set of questions, called TRIAGE, that supports PTs in identifying and discussing medication-related problems during encounters, and provide them with an overview of possible follow-up actions to solve medication-related problems together with the patient, and (2) to perform a proof of principle test to gain insights into the practical potential of TRIAGE in daily pharmacy practice when used with patients collecting blood-pressure-lowering medication.

## 2. Materials and Methods

### 2.1. Development of TRIAGE

TRIAGE was developed based on insight from the literature; input from PTs and pharmacists during focus group discussions; input from the project team (experts in the field of patient-provider communication, pharmacy practice and research); and input from a steering committee consisting of two patient representatives, two PTs, a pharmacist, a pharmaceutical manager and a general practitioner with a special interest in cardiovascular disease management as patients who collected their cardiovascular medication were encountered (see Section 2.2.2.).

#### 2.1.1. Literature

First, existing tools and questions were identified from the literature to explore patients’ experiences with their medication as an inspiration for the TRIAGE question set. The literature was also searched to identify problem domains with medication intake as input for an overview with possible problems and its potential causes to use by PTs (see Table 1).

#### 2.1.2. Input from Steering Committee and Pharmacy Team Members

This first literature-based version of TRIAGE (the question set and overview) was discussed with the steering committee and the adjusted version was then discussed in two separate focus group discussions—one with eight PTs and one with seven pharmacists. Feedback was provided on the number of questions and their formulation, the identified problem domains and how they perceived the feasibility of TRIAGE in daily practice. The refined TRIAGE instrument that was ready to be tested consisted of four open questions on the following topics: use of the medication, side effects, perceived necessity and concerns, and a closing question. Suggestions for follow-up questions per topic were provided. The overview with follow-up actions included ten problem domains (see Table 1). The (lack of) social support is not easily recognized by the PTs, nor can a solution easily be provided. This domain was replaced with “being unable to take certain (auxiliary) substances”, as this was mentioned in the focus groups as being a relevant problem. The most common causes have been identified for each domain, and possible follow-up actions and possible solution were suggested.

#### 2.1.3. Pre-Test

Two PTs from one pharmacy pre-tested TRIAGE for one day, mainly to check whether there were any major obstacles in using TRIAGE in their daily practice. As a result, the formulation of one of the opening questions was slightly adjusted. The final question set is shown in Table 2. The overview with suggested solutions per problem domain can be requested from the corresponding author (MV).

#### 2.1.4. Incorporation in Pharmaceutical Care Program

The TRIAGE question set was then digitalized and incorporated in the “Integrated Pharmaceutical Care Program” of BENU Apotheken, one of the largest pharmacy chains in the Netherlands. With this program, available for all pharmacies belonging to this chain, the pharmacy team can monitor patients’ refill pattern, e.g., if patients collect the medication too early or too late, and it can alert PTs through a pop-up on the computer screen when an action is needed. With the integration of the TRIAGE questions in this program, all participating pharmacies could register their TRIAGE activities in the same way (further details on the procedure in Section 2.2.3). The overview could not be digitalized in this system as it did not fit in the digital format that was already in place. It was therefore provided to PTs as a laminated one-pager to keep at the pharmacy counter.

### 2.2. Evaluation of TRIAGE in Community Pharmacies

#### 2.2.1. Setting

TRIAGE was tested in 10 community pharmacies from the pharmacy chain BENU Apotheken by convenience sampling. The pharmacies were located throughout the Netherlands. In total, 17 PTs participated, two per pharmacy, except for the pharmacies with a small PT-team.

#### 2.2.2. Patients

The PTs conducted TRIAGE conversations with patients collecting blood-pressure-lowering medication, i.e., diuretics, beta-blockers, calcium antagonists, or renin–angiotensin system inhibitors, for themselves. This was either a first refill prescription or a follow-up refill prescription for which it was known that the patient collected the medication too late or too early. This was identified by calculating the percentage of days covered based on the medication the patient collected in the preceding year. The prescription duration in the Netherlands usually is 90 days, so every three months the patient needs to collect medication from the pharmacy. If the patient collected less than 80% or more than 120% of the medication in the preceding year, an alert was given by BENU’s pharmaceutical care program. Patients using blood-pressure-lowering medication were included for the evaluation of TRIAGE to delimit this practice test. In addition, adherence to this type of medication is known to be suboptimal [20], thus, it was expected that PTs would come across adherence problems.

#### 2.2.3. Procedure

All participating PTs followed a two-hour Skype workshop facilitated by two trainers (MV, AL). The PTs either joined one of two evenings at which the workshop was given or viewed the recorded workshop after which they could ask questions during a scheduled telephone appointment with one of the trainers (MV). The workshop aimed to refresh their communication skills and to provide instructions on how to use TRIAGE in their daily practice. An important aspect of TRIAGE is that it is not meant as a checklist for PTs; for instance, it was not required to ask all four questions. TRIAGE aimed to support the PTs in initiating the conversation, after which it gave the PTs room to make the conversation their own by aligning the conversation to their own communication style and with the flow of the conversation.

Over two months (November 2017 and January 2018, leaving December 2017 out of the test due to end-of-year pressure in the pharmacy), PTs conducted TRIAGE conversations. When eligible patients presented themselves in the pharmacy, a pop-up was created to alert PTs to start a TRIAGE conversation. The PTs were then led through the TRIAGE questions. The following aspects were registered in the pharmacy system: (1) whether a problem was identified and, if so, which problem was to be chosen out of the 10 problem domains; (2) which action was taken by the PT (which suggestion for solution); and (3) whether any follow-up actions were necessary, e.g., whether the problem was solved on the spot, a referral to pharmacist or prescriber was necessary, or other actions. After each conversation, PTs were asked to indicate if they asked all four questions, and how long the TRIAGE conversation took.

#### 2.2.4. PTs’ and Patients’ Experiences with TRIAGE

The PTs were asked to complete a short questionnaire before and directly after the two-month study period. The questionnaire included three topics. The first topic assessed the PTs’ ability to identify, discuss and provide solutions for problems in medication use using statements, e.g., “I find it difficult to identify any problems with the patient’s medication use”. Answers could be given on a five-point scale from never to always. The second topic assessed how often the PTs asked patients to share their experiences with the medication also using statements, e.g., “I ask patients for their experiences with the medication at first refill”, with answers to be given on the same five-point scale. The third topic assessed how TRIAGE would help them in their conversations (before), and whether TRIAGE had helped them (after) both by means of an open question. In addition, in-depth telephone interviews were held with one PT per pharmacy to gain further insight into their experiences with working with TRIAGE in their daily practice. After the TRIAGE conversation, PTs provided patients with a one-page written questionnaire to assess their experiences with the conversation they had just had with the PT. Patients were asked to complete this at home and return it to the researchers. PTs had no access to the answers.

## 3. Results

### 3.1. Identified Problems in TRIAGE Conversations

Over two months, 105 TRIAGE conversations were held; 66 with patients collecting a first refill prescription, 39 with patients collecting a follow-up refill prescription. The number of conversations varied between pharmacies mainly due to the size of the pharmacy (see Table 3). Two PTs from two small-sized pharmacies did not have the opportunity to conduct TRIAGE conversations as they indicated that no eligible patients came to their pharmacy during their work shifts (both PTs worked part-time) in the two months TRIAGE was tested.

A medication-related problem was identified in 15 (23%) of first refill prescription encounters and in three (8%) of follow-up refill prescriptions. At first refill encounters (Figure 1), most of the identified problems related to (practical) barriers for the medication intake, followed by the complexity of the medication (regime). All identified problems appear to have been followed by a suitable action, except for a conversation in which the patient expressed fear of side effects. Most of the problems were solved on the spot by the PT, either at the pharmacy counter or in the consultation room. Regarding follow-up refill encounters, two identified problems related to the complexity of the medication regime and one to forgetfulness. Two problems were solved on the spot; for one problem, the PT referred the patient back to the prescriber.

Differences were found in the number of questions asked between first refill and follow-up refill prescription encounters (Table 4). The majority of PTs asked all four questions at first refill encounters, and the PTs generally only asked one of the four questions at the follow-up refill encounters. The length of the conversation was most often less than five minutes for both types of encounters. This was expected as the actions that were taken by the PTs were quite simple ones, e.g., providing a written intake schedule (see Figure 1); these are actions that do not take that much time and need no consult with the pharmacist.

### 3.2. PTs’ Experiences with TRIAGE

Fourteen of the 17 PTs returned the first questionnaire sent out before the study started (response rate 82%), and eight of the 15 PTs (two PTs conducted no TRIAGE-conversation) returned the second questionnaire sent directly after the study ended (response rate 53%). The PTs who responded to the second questionnaire all responded to the first. It appears that PTs tend to perceive the identification of problems and provision of solutions to be less difficult after having worked with TRIAGE in comparison with before. After introducing TRIAGE, PTs asked patients more often about their experiences with the medication (65% versus 50% of PTs at first refill and 50% versus 15% of PTs at follow-up refill). Before the use of TRIAGE, PTs expected the following aspects of TRIAGE to be helpful while conducting conversations with patients: having a structure for the conversation (four PTs), having follow-up questions to ask (four PTs) and asking the right questions (three PTs). Afterwards, six PTs indicated that the open-ended and patient-centered nature of the questions, and having follow-up questions available were most helpful.

One PT per pharmacy was invited for an in-depth interview to further assess experiences with using TRIAGE in their encounters with patients. Seven in-depth interviews were held provided that PTs in two pharmacies did not have the opportunity to perform TRIAGE conversations, and the PT from another pharmacy was unable to schedule an interview. All PTs indicated that the practical questions helped them in starting the conversation about the medication with the patient, and also in keeping this conversation going. Five of the seven PTs explicitly indicated that they identified medication-related problems with TRIAGE that would have gone unnoticed otherwise. They did report that the success of the conversation was also dependent on the attitude of the patient; whether the patient showed an interest in the conversation and whether the patient allowed the time to engage in the conversation. Five PTs did not use the overview with potential solutions to identify problems during the conversation itself, although they thought this overview was complete and useful. A frequently mentioned reason for not using the overview was that, in contrast to the question set, this overview was not digitalized which made it unpractical to look up a solution while talking to the patients, or, in other words, this felt like a disruption to the conversation. Two more experienced PTs mentioned that they could provide solutions from their own experience and therefore did not need to use the overview.

Four PTs noticed differences between the first refill and the follow-up refill TRIAGE conversations. At first refill, patients were more prone to engaging in a conversation about their experiences with the medication they recently started. With patients collecting a follow-up refill, PTs noticed that they did not expect to be asked questions about their medication and that they did not feel the need for a conversation provided that they had been taking the medication for quite some time. Five PTs took the patients to the consultation room instead of having the conversation at the counter. They did this because they thought the patient would more easily share experiences since there was more privacy. They also mentioned another reason for taking the patient to a consultation room: for them to not feel rushed or distracted by other patients in the waiting area.

All but one PT indicated that the TRIAGE conversations could fit in their daily pharmacy practice. The PTs made it clear that this was dependent on the attitude of the patient being open to counseling and the crowdedness of the pharmacy, i.e., having time to start the conversation as well as privacy issues.

### 3.3. Patients’ Experiences with TRIAGE

Out of 105 patients, only 17 patients (16% response rate) returned the questionnaire. Sixteen patients appreciated that the PT asked them about their experiences with the medication and thought it was pleasant to talk about their experiences with the PT. Thirteen of fifteen patients who received advice from the PT found this advice very useful. Furthermore, all 17 patients reported that the PT took time, listened carefully, took the patients seriously, and gave them space to ask questions. However, these experiences are from a very small number of patients.

## 4. Discussion

In close collaboration with patients, PTs and pharmacists, the TRIAGE practical question set was developed for the pharmacy team to identify and discuss possible medication-related problems with patients. TRIAGE also included an overview of suggested follow-up actions for each of ten medication-related problem domains. The proof of principle test showed that TRIAGE was well received by PTs, as they were very positive about the support TRIAGE provided them in having a conversation with patients about their medication use. In addition, a substantial number of medication-related problems was identified in TRIAGE-based conversations, especially during first refill encounters. PTs indicated that they identified problems that would otherwise not have come forward. Moreover, TRIAGE appears to be feasible to use in daily practice.

It was found that at first refill, patients were more open to questions about their experiences with the medication and showed more interest in receiving counseling on their relatively new medication. This is in line with Feifer and colleagues (2010), who showed that patients who have recently started medication appear to have a higher need for information and counseling about their medication [25]. This need was, for example, also seen in patients starting with inhalation treatment [26], and in patients starting with oral blood-glucose-lowering medication [9]. This could also explain why especially at first refill, problems were more often identified than at follow-up refill.

At follow-up refill, patients seemed to be less receptive for a conversation about their medication use. According to the PTs, patients often did not expect these questions. The role of the pharmacy is shifting more to supporting patients in their medication use than only providing patients with a medication supply. However, the pharmacy team still struggles to make their role as a pharmaceutical care provider more visible for patients [9]. Moreover, asking about patients’ experiences and whether they succeed in taking the medication every day (being adherent) is also important for patients who are on treatment for a while. As adherence is a dynamic behavior and varies over time, it is important to stress the need of regularly monitoring patient experiences [27,28]. Targeting specific patient groups for TRIAGE-based conversations at follow-up refill, e.g., patients with polypharmacy or patients who use medication known for heavy side effects, might be more useful.

Privacy issues can hinder an open conversation between the PT and the patient at the pharmacy counter. The patient might not want to share experiences or disclose medication-related problems in a crowded pharmacy. This might have also hindered patients at follow-up refill for whom it was known that they had not taken the medication as prescribed. Some PTs already indicated that they guided patients to the consultation room to start the TRIAGE-based conversation, hoping that this would facilitate disclosure of problems. In this proof of principle study, no distinction could be made between conversations held at the pharmacy counter and those held in the consultation room to investigate whether it actually led to more identification of problems. The privacy issues are acknowledged by community pharmacists, as there is an increase in pharmacists remodeling their pharmacies to provide more privacy, but also to lower the threshold of taking the patient to the consultation room for a conversation (e.g., [29]). This might provide a safer atmosphere for patients to share their experiences and disclose any problems.

TRIAGE provides the pharmacy team with a practical communication tool to open the conversation with the patient and discuss their medication use. However, it is also important to pay attention to the communication skills of the pharmacy team and train these skills to further facilitate medication counseling. Training physicians in their communication skills, for example, has shown to enhance their patients’ medication adherence [30]. Training PTs with an online feedback communication training (COM-MA) has improved patient-centered communication (Vervloet et al. manuscript in preparation). Furthermore, PTs in the Netherlands complete three years of vocational education post-secondary education, during which they learn to inform and counsel patients about their medication. However, it is important to note differences in skills and experience between PTs that might affect the success of identifying problems and discussing solutions in a TRIAGE-based conversation.

### 4.1. Strengths and Limitations

One of the strengths of this study is that TRIAGE was developed in close collaboration with PTs, pharmacist and patients. This ensured that TRIAGE aligns with their needs and preferences and is feasible to use in practice—a prerequisite for a new instrument to be adopted in daily practice. Moreover, as TRIAGE was not advised to be a checklist, it gives room for PTs to make the conversation their own, aligning with their communication style and with the flow of the conversation.

The use of TRIAGE in the conversation with the patients was assessed with self-report by the PTs. To better assess the fidelity of TRIAGE, objective measures need to be used, e.g., by (video) observation of the conversation.

Although it was aimed to assess patients’ experiences with the TRIAGE conversation, the response was too low to be able to show reliable results. This low response might be caused by PTs forgetting to hand out the questionnaire to patients after the conversation. Furthermore, patients needed to remember to complete the questionnaire at home and return it to the researchers. Other ways of engaging patients in the evaluation of TRIAGE need to be sought, since further investigation of patients’ experiences could provide valuable insights into how patients perceive this counseling by PTs, which aspects are most useful for which patients, and how the conversation can further be optimized.

Unfortunately, the overview could not be digitalized, which hindered the use of this aspect of TRIAGE. Optimal integration in the workflow by offering a fully integrated digital solution would have facilitated a more successful implementation.

Ultimately, the aim is to improve patients’ medication use and their adherence. This study was too small to investigate the effect on adherence as it was primarily aimed at testing the TRIAGE instrument for use in the pharmacies’ daily practice. Further research should aim to investigate the effects of patient counseling via TRIAGE on adherence to medication. 

### 4.2. Implications for Practice and Research

Counseling patients in the pharmacy is becoming increasingly important, hence the recently developed guideline ‘Pharmaceutical Consult’ by the Royal Dutch Pharmacist Association (KNMP). TRIAGE facilitates the implementation of this guideline by supporting PTs in starting the conversation with patients about their medication and to further discuss patients’ experiences with the medication. Although TRIAGE was evaluated for conversations about cardiovascular medication, it is a universal instrument which can be used for all types of chronic medication. Pharmacy teams from around the world struggling with having an inviting, open conversation with patients about their medication can benefit from an instrument such as TRIAGE.

This first proof of principle test provided promising results for the application of TRIAGE in daily pharmacy practice. Next steps include investigating how TRIAGE can be improved to better support conversations at follow-up refill encounters and to investigate patient experiences with TRIAGE-based conversations. Furthermore, future research should investigate the effect of TRIAGE-based conversations on patient outcomes, e.g., medication adherence. These future steps have become a possibility now that the TRIAGE question set has been incorporated in the module “First refill support” provided by knowledge center Health Base, which aims to provide knowledge and support to pharmacies in the Netherlands.

## 5. Conclusions

This proof of principle study provided promising results for the application of TRIAGE, a practical communication tool to identify, discuss and solve medication-related problems in daily pharmacy practice from the PT perspective. A substantial number of medication-related problems were identified with TRIAGE during first refill prescriptions encounters, which would have otherwise been left unnoticed according to the PTs. They appreciated TRIAGE as a useful tool for starting the conversation about medication use. Nevertheless, the patient’s perspective on TRIAGE-based conversations is still needed to further evaluate and optimize TRIAGE.

## Figures and Tables

**Figure 1 pharmacy-08-00178-f001:**
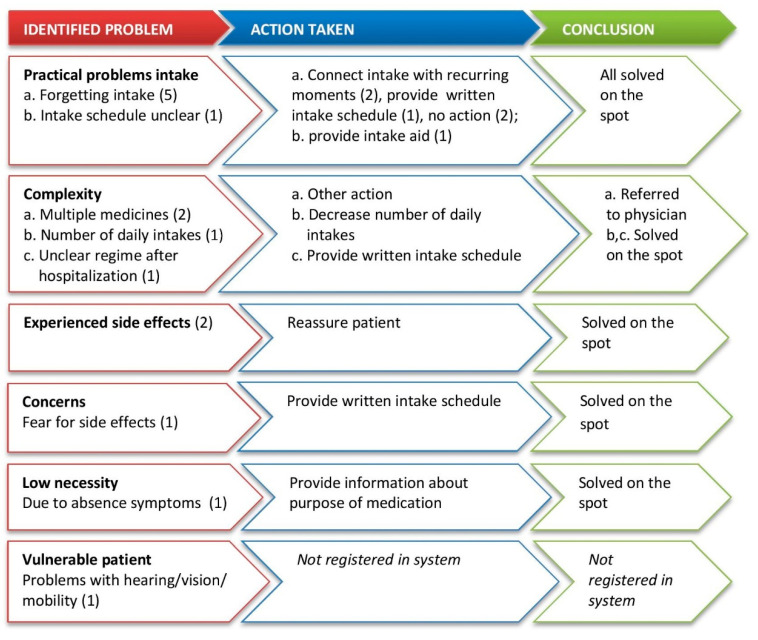
Type of problems identified and follow-up actions at first refill prescription encounters.

**Table 1 pharmacy-08-00178-t001:** Main problem domains identified in the literature.

Problem Domains
1. Practical intake problems;
2. Problems with incorporating intake in daily routine;
3. Complexity of the medication (regime);
4. Experienced side effects;
5. Perceived low necessity of the medication;
6. Concerns about the medication;
7. Knowledge barriers;
8. Costs;
9. Vulnerable patients who are unable to understand or apply information;
10. Social support.

**Table 2 pharmacy-08-00178-t002:** TRIAGE practical question set for first refill and follow-up refill prescription encounters.

Opening Questions	Follow-Up Questions
1. I am curious about your experiences. How have you been taking this medicine lately?	Suggestions for first refill: - Can you tell me a bit more about how you use this medicine? - Is there anything you find troublesome in using this medicine? Suggestions for follow-up refill: - Has anything changed in how you use the medicine? Can you tell me more about that? - To what extent do you manage to take the medicine every day? - How do you make sure you don’t forget to take your medicine?
2. Any medication can also have side effects. How’s that for you? Do you experience side effects of this medicine?	Suggestion for first and follow-up refill: - We know this medicine can cause… (fill in by PT) as a side effect, have you also experienced this?
3. How do you feel about using this medicine (long-term)?	Suggestions for first and follow-up refill: - How important is it to you to use this medicine every day? - Do you have concerns about this medicine? - What do you expect from this medicine? - What would be a reason for you to stop taking this medicine? Suggestion for specific for follow-up refill: - What effect do you experience from the medicine?
4. Which questions do you still have?	Refer to additional, reliable information about the medicine.

**Table 3 pharmacy-08-00178-t003:** Number of TRIAGE conversations held in 10 community pharmacies.

Pharmacy	First Refill	Follow-Up Refill	Total
1	8	7	15
2	7	7	14
3	8	3	11
4	11	12	23
5	13	7	20
6	8	1	9
7	1	0	1
8	4	1	5
9	5	1	6
10	1	0	1
Total	66	39	105

**Table 4 pharmacy-08-00178-t004:** Number of questions asked and length of TRIAGE conversation.

Aspects of the TRIAGE Conversation	First Refill	Follow-Up Refill
Number of questions asked	(n = 37)	(n = 24)
All 4 questions	20 (54%)	6 (25%)
3 questions	7 (19%)	4 (17%)
2 questions	6 (16%)	4 (17%)
1 question	4 (11%)	10 (42%)
Length of TRIAGE conversation		
<5 min	27 (79%)	13 (72%)
5–10 min	6 (18%)	3 (17%)
>10 min	1 (3%)	2 (11%)

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
