# Peer review of "The Development and Proof of Principle Test of TRIAGE: A Practical Question Set to Identify and Discuss Medication-Related Problems in Community Pharmacy"

_pharmacy, 2020, doi:10.3390/pharmacy8040178_

Round 1

Reviewer 1 Report

Thank you for the opportunity to contribute to the peer review process for the original manuscript submission entitled "The development and pilot test of TRIAGE, a practical question set to identify and discuss medication-related problems in community pharmacy (pharmacy-820835)". The authors report the results of their pilot efforts to develop a practical communication tool to support pharmacy technicians in initiating a conversation across the pharmacy counter on medication-related problems. It was concluded from across 10 community pharmacies, 17 technicians whom conducted 105 conversations using the TRIAGE tool that the tool was appreciated and useful to initiate the medication usage conversation.

Reviewer General Comments
The following comments & suggestions are made to assist authors improve the reporting of their work and readability for the Pharmacy audience.

Throughout the manuscript those attending the community pharmacy are referred to as patients; this reviewer recommends that such individuals in the ambulatory care / community setting may be better termed with more empowering language such as customers or clients.

Suggest supplementing to describe what is the reason this is a pilot study; Is it feasibility / proof of concept approach etc? Why is this approach necessary? Reasons for conducting pilot or feasibility studies can fall under four categories: to inform process (e.g., feasibility of recruitment, retention, intervention adherence), to understand resource requirements (e.g., time and budget issues), to inform management (e.g., personnel challenges, data collection or organization), and to advance scientific inquiry (e.g., intervention safety, appropriate dose, potential treatment effect) [Thabane L, Ma J, Chu R, Cheng J, Ismaila A, Rios LP, et al. A tutorial on pilot studies: the what, why and how. BMC Med Res Methodol. 2010;10(1):1–10. http://bmcmedresmethodol.biomedcentral.com/articles/10.1186/1471-2288-10-1]. If feasibility of their TRIAGE approach was the aim to be established then objectives, akin to those of Orsmond and Cohn [Orsmond G, Cohn, ES. The distinctive features of a feasibility study: objectives and guiding questions. OTJR (Thorofare N J) 2015; 35(5): 169-177; https://doi.org/10.1177/1539449215578649] might reasonably be expected to be a feature of the methods and focus of the subsequent reporting:

These five objectives being:

  1. evaluation of recruitment capability and resulting sample characteristics
  2. evaluation and refinement of data collection procedures and outcome measures
  3. evaluation of acceptability and suitability of the study procedures
  4. evaluation of resources and ability to manage and implement the study
  5. preliminary evaluation of participant responses

It is recommended that the authors present in the methods why the pilot study approach was taken & clarify if in fact their aim was to explore feasibility or establish proof of concept and/or to inform the conduct of a planned larger future study. Success against this can then inform the Discussion further. Then page 9, lines 329-333 can be expanded to incorporate more definitively the translation from pilot / feasibility / proof of concept into future research (before one contemplates large-scale implementation...).

Specific Comments

Abstract, page 1 of 11, lines 23, 31 & 32 - suggest amending such that sentences do not start with an abbreviation.

Introduction, page 2, line 80 - suggest amending to utilize previously defined abbreviation 'PT' here

lines 88-9 - suggest review of parenthesis use here such that nested / double enclosure is not using same rounded parethesis.

Materials and Methods, page 3, line 91 - suggest amending to depersonalize

line 94 & 95 - suggest amending to depersonalize

Page 4, line 134 - why 10 pharmacies? Why 17 PTs recruited to participate? Suggest supplementing here to state explicitly that this was a convenience sampling to avoid concerns of selection bias.

line 138 - suggest amending such that sentences do not start with an abbreviation.

line 139 - abbreviation RAS does not appear subsequently in manuscript - so of no utility to define here.

line 142 - suggest review lone/orphaned end/closing parenthesis (after respectively).

line 144 - suggest amending to depersonalize: 'Patients using blood pressure lowering medication were included for...'

line 146 - suggest amending to depersonalize

Page 5, line 159 - suggest amending such that sentences do not start with an abbreviation.

line 167 - suggest amending such that sentences do not start with an abbreviation: 'The PTs were...'

line 171 - why in-depth interviews with 7? How was this number determined? How were the 7 chosen? Why this approach and not a focus group? Were these conveniently hand-picked or volunteers etc?

As there was no sample size calculation or desired power conversation provided, even for a pilot sample there should be some rationale conveyed for the numbers within the sample selected. As part of piloting/feasibility there still needs to be consideration as to whether recruitment was successful for numbers and sample reflection of the wider population, so a desired target and strategy to recruit should still be explicit and available for external validity consideration.

Results, Page 6, line 208 - suggest amending such that sentences do not start with an abbreviation: 'The PTs appeared...'

line 217 - suggest amending to depersonalize.

Page 7, line 247 - suggest amending to depersonalize.

Discussion, line 258 - suggest amending to depersonalize: 'It was found...'

Page 8, line 282 - suggest amending to depersonalize.

line 293 - suggest amending to utilize previously defined abbreviation 'PT' here

line 295 - this reviewer will defer to editorial office direction as to whether inclusion of an-text citation to an as yet unavailable ('in preparation' from the CI) resource is permissible.

line 304 - suggest amending to depersonalize: 'Although it was aimed...'

line 316 - suggest amending to depersonalize.

Conclusion page 9 - suggest that the statements here need to be tempered with additional comments that this is based on a limited selected sample as a pilot exercise, within a cohort having a narrow sliver of medications dispensed (cardiovascular).

It is incongruous to conclude that there are substantial problems identified but then that the tool is suitable for implementation on a daily basis.

The conclusion needs to better reflect the pilot nature of the work and contain statements that address the issues as to success against the reasons to conduct a pilot approach in first instance. Was it feasible, valid, reliable etc etc?? Then it would be reasonable to extend to saying that the TRIAGE has promise from a PT perspective, but that client input needs to be explored further in future trials before widespread implementation can be advocated.

References

#1 - review capitalization of article title

#5 - review italicization of journal title for consistency with rest of list

#7 - review journal title for abbreviating

#9 - review journal title for abbreviating

#12 - review capitalization of article title & journal title for abbreviating

#17 -review book title word spacing (withPatients)

#18 -review capitalization of article title & inclusion of url/doi for accessing

#20 - review capitalization of article title

#22 - review capitalization of article title & italicization of journal title.
Further suggest review from line 413 as it appears a second reference is continuous here rather than uniquely numbered - amend here & review in-text citations for accuracy for subsequent numbering
If Barnett et al is to be included - review italicization of journal title

line 421 - Kooy et al reference not numbered (?25 or 26 depending on inclusion of Barnett et al). Also review italicization of journal title.
Further, there is no in-text citation for [25] - so suggest review all in-text citations beyond #22 for correct alignment to reference list

Reviewer 2 Report

This manuscript refers an interesting study that can have practical application in the work of community pharmacy.

I think that in the introduction the authors should stress the fact that there is a need of new orientations and increasing implication of pharmaceutical care in community pharmacy.

I agree with them that sometimes is difficult the contact with the patient/costumer, and it is a barrier that should be overcome in order to have the possibility to have more influence in the patient treatment and management.

Other things to consider:

  • Results: line 160, the authors state that the pharmacist register his/her activity in the pharmacy system. All the pharmacies that participated in the study had the same software? How was made possible to homogenise the recollection of the activities?
  • Figure 1 should be more explained in the text as there are some aspects that are not clear to the reader. For example : concerns (fear of side effects) how this can be solved providing only written intake schedule? Side effect do not depend on this, at least exclusively.

In the second box of identified problems I think there is an error, the title (complexity) is shown with an “a” as a subsection.

Reviewer 3 Report

Thank you for submitting your interesting project.  Patient counseling is always an important topic in pharmacy.

I have a couple of questions/concerns.

Methods:  What role does the pharmacist play in this project?  The regulations may differ in your country, but in my country a pharmacy technician cannot make any recommendations to a patient.  Those must be handled by a licensed pharmacist.  I did check the Royal Dutch Pharmacist Association and it states that one of the primary roles of a pharmacist is to provide counseling. Where is the role of the pharmacist in this project?

I am more than concerned that a pharmacy technician would answer questions on side effects based upon their own personal experience.  How can a pharmacy technician provide a recommendation on reducing the daily number of intakes (Figure 1).  This is beyond the scope of a pharmacy technician.  If these are part of the normal functions of a pharmacy technician then maybe an explanation of the schooling and extensive training they must undergo to be a pharmacy technician should be explained and the regulations that allow this practice.  A clear explanation of the role of the pharmacist in this process should be done.  It is one thing to start the communication with this tool but another to actually make recommendations without a pharmacist input.

Explain how providing a written intake schedule helped with a fear of a side effect.  What was the side effect in question that this was sufficient.

Table 1 I am concerned that conversations with patients that needed an intervention was completed in less than 5 minutes.  Was this the amount of time the technician spoke with them before the pharmacist took over?

Reference #22 has two references on one line.  I believe reference #23 should be the  Barnett reference and the following re-numbered.

I really like the idea of your tool but there must be a clearer picture of the role of the pharmacist in the recommendations made, as this would be beyond the scope of a technician.

Round 2

Reviewer 1 Report

Thank you again for the chance to contribute to the peer review process for the revised submission of the manuscript entitled "The development and pilot test of TRIAGE, a practical question set to identify and discuss medication-related problems in community pharmacy (pharmacy-820835 )". The authors have provided a comprehensive response outlining their amendments arising from the editor's and peer reviewer's recommendations of the latest review cycle, and the manuscript has evolved positively via this review process.

The following minor suggested amendments are proposed as part of a final submission / proof review process:

Page 2, line 47 - suggest utilize this opportunity to define the abbreviation '...pharmacy technicians (PTs) and...the PTs. Pharmacy...'

then in line 83 amend to just use the abbreviation here '...that PTs ask...'

Page 5, line 180 - suggest correcting spacing to '...with one PT per...'

Page 6, line 214 - suggest amending punctuation to '...(see Figure 1); these are...'

Page 7, line 231 - suggest correct spacing to '...assess experiences...'

Page 9, line 359 - suggest utilizing previously defined abbreviation here '...from the PT perspective.'

page 10, line 361- suggest utilizing previously defined abbreviation here '...to the PTs. They...'

Author Response

Authors’ response to comments made by reviewer 1 in the second round of revisions

Thank you again for the chance to contribute to the peer review process for the revised submission of the manuscript entitled "The development and pilot test of TRIAGE, a practical question set to identify and discuss medication-related problems in community pharmacy (pharmacy-820835)". The authors have provided a comprehensive response outlining their amendments arising from the editor's and peer reviewer's recommendations of the latest review cycle, and the manuscript has evolved positively via this review process.

Authors’ response: Thank you very much for this compliment.

The following minor suggested amendments are proposed as part of a final submission / proof review process:

  • Page 2, line 47 - suggest utilize this opportunity to define the abbreviation '...pharmacy technicians (PTs) and...the PTs. Pharmacy...'
  • then in line 83 amend to just use the abbreviation here '...that PTs ask...'
  • Page 5, line 180 - suggest correcting spacing to '...with one PT per...'
  • Page 6, line 214 - suggest amending punctuation to '...(see Figure 1); these are...'
  • Page 7, line 231 - suggest correct spacing to '...assess experiences...'
  • Page 9, line 359 - suggest utilizing previously defined abbreviation here '...from the PT perspective.'
  • page 10, line 361- suggest utilizing previously defined abbreviation here '...to the PTs. They...'

Authors’ response: Thank you very much for these suggestions. We have incorporated all suggestions in the revised manuscript.

Reviewer 3 Report

Thank you for your re-submission. Please look at line 200 on page 6. Is the word supposed to be "except"? Line 231 there is a spacing issues with words that needs to be corrected. Well done!

Author Response

Authors’ response to comments made by reviewer 3 in the second round of revisions

Thank you for your re-submission. Please look at line 200 on page 6. Is the word supposed to be "except"? Line 231 there is a spacing issues with words that needs to be corrected. Well done!

Authors’ response: First of all, thank you very much for the compliment. And thank you for noticing the typographical error in the word “expect”, this indeed should be “except”, we changed this. We have reviewed several sentences and corrected spacing issues (also mentioned by reviewer 1).

Finally, a native speaker has checked the English grammar and style.